# Different Weed Managements Influence the Seasonal Floristic Composition in a Super High-Density Olive Orchard

**DOI:** 10.3390/plants12162921

**Published:** 2023-08-11

**Authors:** Stefano Popolizio, Gaetano Alessandro Vivaldi, Salvatore Camposeo

**Affiliations:** Department of Soil, Plant and Food Science, University of Bari Aldo Moro, Via Amendola 165/a, 70126 Bari, Italy; stefano.popolizio@uniba.it (S.P.); gaetano.vivaldi@uniba.it (G.A.V.)

**Keywords:** weed control, soil management, weed population, chemicals, mowing, mulching, seed bank

## Abstract

Weed management is not yet environmentally, agronomically, economically and socially sustainable in olive orchards. It is necessary to study appropriate integrated weed management systems (IWMSs) based on the knowledge of weed population and effects of weeding practices over time. The aim of this study was to evaluate the effects of different weed managements on seasonal floristic composition of a super high-density olive orchard, also exploiting the essential principles of an IWMS. Five weeding techniques were compared: chemical control (CHI), mowing (MEC), plastic (nonwoven tissue, TNT and polyethylene, PEN) and organic (with de-oiled olive pomace, DOP) mulching. Weed monitoring was carried out on six dates in a three-year period. The infestation of each of the main 18 weed species recorded (%) and the total infestation (%) on each monitoring date were determined. Results underlined that all weeding practices investigated in this multi-year study affected the floristic composition, weed characteristics (hemicryptophytes, cryptophytes and therophytes) and seed bank. TNT and PEN were the most effective methods for weed management. Particularly, total infestation coefficient was significantly lowest when plots were managed with TNT (13.91%) and PEN (14.38%) and highest for MEC (141.29%). However, DOP also significantly reduced infestation compared to CHI and MEC. Therefore, DOP could constitute an excellent strategy for weed management in super high-density olive groves, since it also has the possibility of distributing mulching materials in a mechanized way in field and can result in improvement of soil fertility and the possibility of valorizing waste. Further studies should be carried out to investigate the mechanism of action (physical and allelochemical) of de-oiled pomace or other organic agro-industrial materials and the recovery time of these mulching materials in super high-density olive orchards.

## 1. Introduction

Olive (*Olea europaea* L.) is a fruit tree crop widespread around the Mediterranean area. In Italy, olive trees are grown mostly in traditional orchard systems, with tree density varying from 100 to 200 trees per hectare [1]. However, olive cultivation is moving towards intensification and, in the last 10 years, in Italy, more than 5000 hectares of super high-density olive plantations have been established. These new cropping systems are characterized by a highdensity of trees (>1200 trees per hectare) as well as by interesting advantages mainly due to total mechanization of cultivation practices [2].

Weed control is considered one of the main problems for the environmental, agronomic, economic and social sustainability of these cultivation systems [3] because weed species compete with trees for water and nutrients [4,5], reducing fruit yield and quality [6]. Moreover, weeds are often hosts of vectors of important diseases, such as *Xylella fastidiosa* in the case of the olive tree [7]. Furthermore, the main techniques of weeding in olive groves are mechanical (soil tillage) and chemical (with pre- and post-emergence herbicides) control. Soil tillage, in addition to destroying weeds, temporarily improves rainfall infiltration and water distribution in the soil profile and incorporates fertilizer into the soil [8,9,10]. However, this practice also results in degradation of the soil structure over time, which can reduce water infiltration rate, accelerating runoff and the erosion process and reducing water availability in the rhizosphere [10]. On the other hand, although the effectiveness of chemical herbicides is very high, their persistent use can result in an impoverished soil quality, in a decrease in orchard biodiversity and in more tolerance or even resistance of weeds to chemical herbicides [11,12,13,14]. Therefore, particularly in olive groves of the semi-arid Mediterranean area, farmers are seeking sustainable weeding practices (single or integrated) that are alternatives to mechanical and chemical control.

The concept of weed control has been replaced by that of Integrated Weed Management System (IWMS) [15], defined as “a holistic approach to weed management that integrates different methods of weed control to provide the crop with an advantage over weeds” [16]. Furthermore, IWMS consists of optimizing all knowledge available in the field of agronomy, physiology, plant cultivation, agricultural ecology, technology and weed biology [17]. The main objective is to minimize the harmfulness of weeds in the short and longterm by following an ecologically, economically and agronomically sustainable approach. The IWMS in orchards is essentially based on the three following general principles: (i) the reduction or, when possible, the elimination of soil tillage; (ii) the diversification of weeding strategies over time (use of different agronomic techniques in the year) and space (in the inter-rows and on planting rows); (iii) seasonal weed control (namely, when the weeds can compete with the trees for water and nutrients) [4,5]. In addition, proper weed management in orchards should take into account the geographical area (arid, semi-arid, dry sub-humid, sub-humid, etc.), the age of planting (since competition for water and nutrients is much greater in the early critical stages of tree growth [18,19] due to the low root density of trees compared to weeds [4]) and the fertility of soil [20].

The inter-row zone is a clear area where the tractor and machinery pass; therefore, strategies to provide soil physical protection, to reduce erosion and to minimize compaction should be in place. Several studies have shown the multiple functions of cover crops in olive orchards in the Mediterranean area [21,22,23,24,25,26,27]. In addition, cover crops lead to increased biodiversity, which is essential for orchards’ sustainability [20].

Conversely, a serious challenge for farmers is managing weeds in tree rows. Mulching is an agronomic practice which consists of placing organic or inorganic covering materials on the soil surface of tree rows to minimize moisture losses and weed population and to enhance crop yield and environmental sustainability [28,29,30]. Larsson [31], Dale [32] and Nilsen et al. [33] also observed the positive effects of mulching on fruit size and yield. The inorganic mulching, made with plastic films, reduces weed seed germination by shading and physically blocking emergence from soil [34,35], although with different results depending on the types of plastic materials (by color, opacity, density, etc.) used. Furthermore, mulching with plastic films can also minimize weed management costs as a result of the low cost and long-term efficiency of plastic [19] and savings of 75% on irrigation water use [36]. Polyethylene and polypropylene are cheap and the most widely used materials in agriculture, highly resistant to physical and chemical factors and slowly biodegradable due to inactive groups that are resistant to biological attacks. Even nonwovens have a high resistance to physical, chemical and biological attacks [37]. Therefore, inorganic mulch can provide a long-term weed management. However, inorganic mulching also has disadvantages [38], including practical problems for farmers, such as labor-intensive tasks to cover the soil surface of orchards due to the lack of specialized agricultural machinery for applying and maintaining inorganic mulches in orchards [39]. In most cases, despite the higher yields obtained when the rows of vines were managed with inorganic mulch compared to a traditional cover crop, lower and higher net gains were recorded, respectively, for the two management systems [40] due to the high labor costs to manage plastic films. However, Hegazi [41] and Coventry et al. [42] demonstrated, respectively, that the plastic mulches are useful in extreme climatic situations, namely for the essential advancing of ripening when the season is cool and short and for conserving water when temperatures are very hot and evaporation is high.

On the other hand, organic mulching involves the use of crop residues derived from agronomic practices or agro-industrial processes. Organic mulches simultaneously perform a physical and allelochemical action on seed germination [35,43,44]. Furthermore, the use of organic mulches in cropping systems has positive effects on soil fertility (physical, chemical and biological) and crop physiology [35,45] and should be considered as more sustainable in terms of environmental, agronomical and economical sustainability, especially in areas where they are produced in large quantities and in short periods of time and because the materials would otherwise be considered as waste. In this regard, the investigation by Larsson et al. [31] showed that organic mulching, compared to plastic mulching, increases fruit size. Granatstein and Mullinix [46] and Ingels et al. [47] concluded that wood-chip mulches are the best organic mulches for effective control of weeds in an orchard and a single application of these materials can also save on irrigation water by over 20% [18]. However, its purchase and use cost is high [48,49]. Therefore, alternative organic mulches should be tested. Ferrara et al. [50] evaluated the effects of exhausted olive pomace in a vineyard.

There are many reported studies on the effects of different soil management systems on yields and fruit quality in olive orchards as a consequence of weed control, reducing moisture losses, increasing fertility, etc. However, an appropriate integrated weed management system should also require knowledge of weed populations and effects of weeding practices on populations over time [51]. In the literature, there are few investigations of this topic, especially in super high-density olive groves. The aim of this study was to evaluate the effects of five different weed managements on floristic composition of an adult super high-density olive grove over time through six monitoring campaigns carried out over three years.

## 2. Results

### 2.1. Exploratory Data Analysis

Eighteen main weed species, with hemicryptophytes, cryptophytes and therophytes characteristics, were detected in our three-year study (Table 1).

The species detected in two or more monitoring dates were *Calendula officinalis* L., *Convolvulus arvensis* L., *Conyza canadensis* L., *Diplotaxis erucoides* L., *Medicago polymorpha* L. and *Sonchus oleraceus* L. Non-homogeneous variances were found for *Convolvulus arvensis* L., *Conyza canadensis* L., *Diplotaxis erucoides* L. and *Medicago polymorpha* L. Thus, infestation data (%) for these weed species were transformed. Conversely, the original data were used in the statistical analysis for total infestation coefficient (TI, %) and for infestation (%) of *Calendula officinalis* L. and *Sonchus oleraceus* L. Combined ANOVA showed that weed management, monitoring date and their interaction significantly affected all variables (Table 2). When weed species were not present, infestation was considered 0.0%.

### 2.2. Weed Infestation Results of Different Weeding Treatments on a Single or More Monitoring Dates

Table 3 shows the differences between weed managements and monitoring dates, derived from combined ANOVA and Duncan tests. Results of the interaction between weed management and monitoring date are shown in Figure 1.

Table 4 shows results of differences between weeding practices for weed species observed on a single monitoring date.

No significant differences were recorded for all variables studied between TNT and PEN (Table 3 and Table 4).

Total infestation coefficient (TI) was significantly lowest when plots were managed with inorganic mulch (13.91% and 14.38% for TNT and PEN, respectively) and highest with mowing (MEC, 141.29%). Furthermore, the total infestation of DOP (46.69%) was significantly lower than CHI (74.62%) (Table 3).

Infestation in DOP was significantly lower or equal to CHI for all tested variables in our investigation, except for *Medicago polymorpha* L. (higher infestation in DOP than CHI) (Table 3 and Table 4). The Dunnett test, applied for weed species detected on a single monitoring date, also returned statistically lower or equal infestation values when organic (DOP) or inorganic (TNT and PEN) mulching was compared with chemical (CHI) and mechanical (MEC) weeding on all monitoring dates, except November 2009, when only inorganic mulching with TNT and PEN was more effective than MEC for all weed species recorded in this monitoring date (Table 4).In May 2007, *Amaranthus retroflexus* L., *Anagallis arvensis* L., *Chenopodium album* L. and *Setaria verticillata* L. infestation was not different for CHI and MEC. In addition, for *Stellaria media* L., no differences were observed between CHI, TNT, PEN and DOP and between CHI and MEC. However, TNT, PEN and DOP completely inhibited infestation of this species (Table 4).

Mulching with DOP was not significantly different from TNT and PEN for all species, except *Avena sterilis* L., *Sonchus oleraceus* L., *Medicago polymorpha* L. and *Lolium rigidum* Gaud. (Table 3 and Table 4).

The Duncan’s test showed the highest infestation of *Convolvulus arvensis* L., *Conyza canadensis* L., *Sonchus oleraceus* L., *Diplotaxis erucoides* L. *Calendula officinalis* L. and *Medicago polymorpha* L. when plots were managed with MEC (Table 3).

Total and *Conyza canadensis* L. infestation was significantly higher on the first (May 2007) and last (November 2009) monitoring date. In addition, for *Conyza canadensis* L., the lowest values were recorded on October 2007 (6.56%). For *Convolvulus arvensis* L., the infestation was null in May 2007 and increased significantly up to June 2009 and disappeared again in November 2009. Infestation of *Sonchus oleraceus* L. in June 2009, November 2009 and October 2008 was not significantly different from May 2007. *Diplotaxis erucoides* L. had significantly lower and close to zero infestation values on all monitoring dates, except in October 2007. *Calendula officinalis* L. infestation was null (0.00%) in the May and June months of the three years. Increasing values of *Medicago polymorpha* L. infestation were observed on the first three monitoring dates. Then, in October 2008 and June 2009, the infestation almost or totally disappeared (0.39% and 0.00% on October 2008 and June 2009, respectively) and finally increase significantly in November 2009 (Table 3).

The interaction results of combined ANOVA (Figure 1) highlight that, on all monitoring dates, both inorganic and organic mulching were significantly more efficient than mowing. Inorganic mulching was also more or equally effective compared to chemical weed control. Total infestation (TI) measured for plots managed with DOP compared to TNT and PEN was statistically equal in all periods except in May 2008 (higher TI values in DOP than TNT and PEN), June 2009 and November 2009 when there were higher TI values in DOP than TNT and in DOP than TNT and PEN, respectively. When chemical weeding was used, total infestation had a decreasing trend over time up to June 2009; on the other hand, a non-significant slight increase in November 2009 compared to May 2008, October 2008 and June 2009 was recorded. Mulching with TNT and PEN returned lower TI values on all monitoring dates, except the last one (November 2009). Organic mulching (DOP) returned to significantly lower levels of infestation in May and October 2007 and in October 2008. Conversely, in May 2008, June 2009 and November 2009, the highest values were recorded, with non-significant differences between May 2008 and November 2009 and between May 2008 and June 2009 (Figure 1).

In May 2007, no significant differences between MEC and CHI were found for *Conyza canadensis* L., *Sonchus oleraceus* L., *Diplotaxis erucoides* L., *Calendula officinalis* L. and *Medicago polymorpha* L. The greatest infestation of *Conyza canadensis* L. was mostly observed in the periods June 2009 and October 2008 in plots managed with MEC. On the first two monitoring dates, the *Conyza canadensis* L. infestation was significantly lower in TNT, PEN and DOP plots than in CHI and MEC but they were not different from each other; while, in May 2008, October 2009 and June 2009, no differences were recorded between inorganic mulch (TNT and PEN), organic mulch (DOP) and chemical weed control (CHI). CHI, PEN and DOP treatments maintained the same efficacy on *Conyza canadensis* L. throughout the investigation. However, in June 2009, the infestation of this species in DOP was significantly higher than the first monitoring date. On the other hand, the TNT treatment significantly reduced its efficiency compared with *Conyza canadensis* L. (increased infestation) in the third year (2009) (Figure 1).

Concerning *Sonchus oleraceus* L., in general, infestation of TNT and PEN plots was statistically the same in all monitoring dates. On the other hand, *Sonchus oleraceus* L. in CHI was significantly higher in the first year of the investigation than in the others. Conversely, it was lower in MEC on the last monitoring date compared to the other previous ones. In particular, the CHI treatment was significantly higher than mulching treatments in the first two monitoring dates and the same in the others, except for May 2008 and June 2009, when CHI was significantly higher than and equal to DOP, respectively. Mowing (MEC) for *Sonchus oleraceus* L. always had higher levels of infestation compared to mulching in all periods, except November 2009 (when all treatments showed equal results). Furthermore, in October 2007, as on May 2007, infestation in MEC and CHI had the same infestation of *Sonchus oleraceus* L., while, in June 2009, MEC was equal to DOP.

The interaction between weed management and sampling date showed no significant differences for *Diplotaxis erucoides* L., except in October 2007, when CHI had higher infestation values than MEC, TNT, PEN and DOP.

*Calendula officinalis* L. had similar behavior to *Diplotaxis erucoides* L. Variations concerned the monitoring dates October 2007, when no significant differences were observed between CHI and MEC, and November 2009, when infestation in MEC was higher than the other treatments (which were equal to each other) (Figure 1).

TNT and PEN maintained the same efficacy on *Medicago polymorpha* L. throughout investigation period and showed no differences between them. Similar behavior was observed for CHI, with the exception of October 2007 when a significantly higher *Medicago polymorpha* L. infestation was recorded compared to other monitoring dates. In October 2007, higher values of *Medicago polymorpha* L. were recorded in CHI than TNT, PEN and DOP but no differences were observed between CHI and MEC. Furthermore, higher *Medicago polymorpha* L. values in DOP were recorded than all other treatments and only compared to CHI, TNT and PEN on May 2008 and November 2009, respectively.

*Convolvulus arvensis* L. infestation was zero on the first and last sampling date and for all weed management techniques (Figure 1). On October 2007, May 2008 and October 2008, no significant differences were recorded both between mulching techniques (organic and inorganic) and with chemical weeding. On the same monitoring dates, the numbers of this weed in the MEC plots was significantly greater than other treatments. In particular, *Convolvulus arvensis* L. was statistically the same in October 2007, May 2008 and October 2008 when MEC was used; then, it increased significantly in June 2009 and disappeared in November 2009. In June 2009, DOP had a higher level of *Convolvulus arvensis* L. infestation than CHI, TNT and PEN.

In general, Table 4 and Figure 1 show less and less evident differences over time in all plots (MEC, CHI, TNT, PEN and DOP) for almost all species detected (*Avena sterilis* L., *Hordeum murinum* L., *Lolium rigidum* Gaud., *Senecio vulgaris* L., *Stellaria media* L., *Convolvulus arvensis* L., *Conyza canadensis* L., *Sonchus oleraceus* L., *Diplotaxis erucoides* L. and *Calendula officinalis* L.).

## 3. Discussion

The increase in tree density as well as the intensification of external inputs (e.g., fertilizers, herbicides, pesticides, etc.) and the technical improvements (irrigation and other mechanized management practices) have changed olive-growing systems [52]. Therefore, new integrated weed management practices suitable for these new olive cropping systems need to be studied. Appropriate weed management is vital in olive orchards to minimize weed competition, assuring quality yields and supporting weed biodiversity [20,53,54]. Fruit trees are poor competitors because of their low root density per unit of soil compared to weeds [4]. Therefore, integration of alternative weed management techniques into innovative cropping systems such as super high-density orchards, following the new definition of integrated weed management [15], can represent an agronomically, environmentally and economically sustainable alternative for farmers. This study aimed to evaluate the effect of different weed management methods in a super high-density olive orchard, exploiting the essential principles of an IWMS, and the direct effects of five different weeding techniques, namely chemical control (CHI), mowing (MEC), inorganic (TNT and PEN) and organic (DOP) mulching. However, the effectiveness of the different weeding techniques can vary over time due to a series of causes (e.g., resistance to herbicides, degradation of mulching materials, composition of weed population according to the characteristics of weed species, etc.). Therefore, the evolution of weed populations and the short- and long-term effects of weeding techniques were investigated in our study. The monitoring dates were chosen according to the ecophysiological traits of the olive grove and, above all, their critical period for weed competition [55,56,57].

All the weeding practices used in this multi-year study affected the evolution of weed populations: floristic composition recorded on the last monitoring date (November 2009) is completely different from that observed on the first date (May 2007) (Table 1).

In our study, inorganic plastic (TNT and PEN) and organic (DOP) mulching seems to have been the most effective weeding practice, in agreement with the literature by Hegazi [41], Coventry et al. [42], Schonbeck [34], Goswami and O’Haire [37], Hammermeister [19] and Fracchiolla et al. [35]. This consideration derives from the following results:i.The total infestation (TI) and the infestation of all species detected on each monitoring date for TNT, PEN and DOP was less or equal to chemical (CHI) and mechanical (MEC) weeding.ii.The prevalence of therophytes species recorded (Table 1) (i.e., annual plants surviving harsh conditions as seeds, therefore, completing their life cycle from seed to seed and dying) and the prevalence of weeding techniques, namely inorganic and organic mulching, that physically impede seed germination as a consequence of shading and soil temperature variation [34,35,58]. Furthermore, organic mulching performs an allelochemical action on seed germination [34,43,44,45] mainly due to the bioactive fraction of organic materials, such as biophenols in the mulch [35,44,45,59,60].iii.Highest total infestation, weed characteristics and lowest infestation of all species when plots were managed with TNT, PEN and DOP on May 2007 is attributable to the deposition of seeds in the soil at the beginning of investigation: one of the most important sources of weeds is the seed bank of soil [61]; therefore, of the five treatments performed about two months before the first monitoring date (March 2007), only TNT, PEN and DOP significantly reduced the infestation of all therophytes species thanks to their mechanism of action. In fact, the inorganic mulching acts directly on the seeds, hindering germination [34,35,58]. These results are also supported by the absence of differences between CHI and MEC for all therophytes species recorded in May 2007 (*Amaranthus retroflexus* L., *Anagallis arvensis* L., *Chenopodium album* L., *Setaria verticillata* L., *Conyza canadensis* L., *Sonchus oleraceus* L., *Diplotaxis erucoides* L., *Calendula officinalis* L. and *Medicago polymorpha* L) (Table 4 and Figure 1). In addition, although no differences in *Stellaria media* L. were detected when TNT, PEN, DOP and CHI were compared, the inorganic mulch completely controlled its infestation. Therefore, the cause of the lower TI on May 2007 cannot be attributed to CHI and MEC because both do not act at the seed stage but when the plant has already developed, without affecting with the seed bank. The commercial product used in CHI was glyphosate, namely a systemic post-emergence herbicide that is directly absorbed by leaves and stems of plants and translocated throughout the plant. In general, results regarding the interaction of combined ANOVA for all variables also highlighted that the repeated use of glyphosate did not induce resistance by the weed species but had a cumulative effect reducing significantly or annulling total and individual species infestation over time. However, we conducted a three-year study; thus, we expect that intensive and repeated use of the same herbicides over a long period of time lead to the emergence of herbicide-resistant weeds [11,12,13,14]. For glyphosate, resistance was observed after its repeated use for 10–15 years [62].iv.TI was also the same in May 2007 and November 2009 and different from the other monitoring dates. Furthermore, the differences between inorganic mulch and other treatments were less and less evident over the years until they became, for many species and, in particular, for the therophytes ones, statistically the same at the last monitoring date (*Avena sterilis* L., *Hordeum murinum* L., *Lolium rigidum* Gaud., *Senecio vulgaris* L., *Stellaria media* L., *Convolvulus arvensis* L., *Conyza canadensis* L., *Sonchus oleraceus* L., *Diplotaxis erucoides* L. and *Calendula officinalis* L.). These results could be due to the degradation of organic and inorganic materials. In a two-year study (2007–2008) carried out in the same experimental site by Camposeo and Vivaldi [63], the short-term effects of the same treatments were evaluated for some physico-chemical parameters of the soil and physiological characteristics of the superhigh-density olive grove. This study showed that, in 2008, 30–40% surface of the TNT and 10–20% of the PEN plots showed mechanical damage. Conversely, in 2008, DOP was still in its solid state [63]. Furthermore, TI and the infestation of all species in DOP were not significantly different compared to TNT and PEN, especially on the first monitoring dates. Therefore, the greater degradation degree of TNT and PEN compared to DOP observed by Camposeo and Vivaldi [63] and the greater efficacy of the inorganic materials observed in this study would lead us to hypothesize a predominantly allelochemical action by DOP in the early years of the investigation and, subsequently, a predominantly mechanical one equal with TNT and PEN degraded to 30–40% and 10–20%, respectively. This hypothesis is supported both (a) by Meftah et al. [64] who observed that phenols, contained in the byproducts derived from milled olives, once applied to the soil, tend to migrate over 120 cm soil under arid climatic conditions depending on the granulometry and permeability of soil; and (b) by statistically equal or higher levels of species infestation in DOP compared to TNT and PEN on some monitoring dates from 2008 (*Hirschfeldia incana* L.) onwards and, above all, in the last year of our investigation for the species *Lactuca serriola* L., *Avena sterilis* L., *Hordeum murinum* L., *Lolium rigidum* Gaud., *Senecio vulgaris* L. and *Stellaria media* L.v.When MEC was used, *Convolvulus arvensis* L. infestation was not statistically different over time but it was always significantly higher than TNT, PEN and DOP, except on the first and last monitoring date. *Convolvulus arvensis* L. is the only cryptophytic species recorded in our study, which has underground reserve organs such as rhizomes. Mulching and chemical weeding were consistently effective over the years by acting on the rhizome. Although glyphosate is absorbed through the leaves, it is systemically translocated throughout the plant. By contrast, mowing is limited to eliminating the above ground parts of plants without affecting below ground organs. However, the repeated use of mowing over three years proved to be a valid technique for controlling perennial herbaceous species, as shown by the zero infestation of *Convolvulus arvensis* L. in November 2009, which was statistically not different from other weeding techniques.

## 4. Materials and Methods

### 4.1. Experimental Site

This three-year field study was carried out at the experimental farm of the Department of Soil, Plant and Food Sciences, University of Bari, located in Valenzano (Southern Italy, 41° 01 N; 16° 45 E; 110 m a.s.l.). According to USDA classification, the soil texture is sandy-clay-loam, with 63.0%, 21.0% and 16.0% of sand, clay and silt, respectively. The area is characterized by a typical Mediterranean climate with a long-term average annual rainfall of 560 mm, mostly concentrated during the autumn–winter months (long-term average annual rainfall of 560 mm), and a long-term average annual temperature of 15.6 °C.

Three different cultivars of olive trees were planted in spring 2006: Arbequina, Arbosana and Koroneiki. The trees (self-rooted) were trained according to the central leader system and spaced 4.0 m × 1.5 m (1667 trees ha^−1^) with a North–South row orientation, according to the super high-density planting system. Routine cultural practices were set up.

From 2007, five different row soil management methods were compared to evaluate their effects on weed population: mulching with de-oiled pomace (DOP), mulching with nonwoven tissue (TNT) and polyethylene (PEN), chemical weeding (CHI) and mechanical weeding (MEC) (Figure 2). The treatments were compared in a split-plot design with 3 replications, arranging the 3 main plots and the 5 treatments in the subplots. Each subplot, 10.5 m long, consisted of 7 plants.

The TNT, PEN and DOP were lying under the trees along the row, 1.05 m width. The DOP was applied at 3 cm thick by distributing 165 kg per plot, equivalent to 0.33 m^3^ of waste and 15 kg m^−2^ of basis weight. The mulching materials were applied in the middle of March 2007 after the row weeds had been removed by mowing and before regrowth occurred.

The chemical plots (CHI) were treated in March, July and October of each year with the systemic post-emergence herbicide, glyphosate, N-(phosphonometil)glycine, C_3_H_8_NO_5_P, at 2.5 L ha^−1^ of commercial product.

The MEC plots were periodically mowed from April to September and were covered in vegetation from October to March of each year.

### 4.2. Mulching Materials Characteristics

The DOP used was the waste derived from the separation of pomace oil through organic solvent by an oil extraction plant located in Bari. DOP had a bulk density of 500 kg m^−3^. The TNT was a polypropylene film green in color (basis weight, 90 g m^−2^). The PEN was black in color with the same basis weight as TNT (90 g m^−2^). More details are reported in Camposeo and Vivaldi [63].

### 4.3. Weeding Monitoring and Survey of Weed Infestation Method

Weed monitoring was carried out on each plot in three different years (2007, 2008 and 2009) and at six monitoring dates (two for each year): May 2007, October 2007, May 2008, October 2008, June 2009 and November 2009. Therefore, weed monitoring was performed after CHI and MEC treatments of each year. The floristic composition assessment was carried out by means of a transept (50 cm × 50 cm) at 7 points randomly on each replicate, according to the abundance-dominance method [65]. It considers that the same can be occupied by either a few larger plants or by numerous small plants. According to their abundance, for each species recorded, a different index is visually assigned. The discrete field data were transformed in richness (total number of species): the midpoints of each cover interval were used to transform the discrete scale into continuous values and to treat the data algebraically [66]. Criteria for assigning the discrete cover-abundance index and the relative midpoints used for their transformation are reported in Fracchiolla et al. [67]. The numerical information obtained was the infestation of each of the main weed species recorded and the total infestation (TI) on each monitoring date. TI was the sum of all weeds (not only the main ones); thus, its value can exceed 100%. The biological spectrum was also determined according to Raunkiaer’s plant life forms, namely as species belonging to Therophytes, Hemicryptophytes and Geophytes [68].

### 4.4. Statistical Analysis

In this study, the effect of weeding treatments over the six monitoring dates was investigated. Therefore, the cultivars were considered as pseudo-replicates within plots (corresponding to treatments). Data on weed infestation (TI and weed species) were tested for normality using the Shapiro–Wilk test. For weed species recorded in two or more monitoring dates and for total infestation, a nested analysis of variance (ANOVA) was separately conducted for each date. The homogeneity of the variances across the monitoring dates was verified through the F test. It is a condition for valid analysis of variance for combining the data from a series of experiments [69]. When the variances were homogeneous, a combined analysis of variance (combined ANOVA) of data was used. Conversely, when the variance was heterogeneous, a weighted least square (WLS) analysis was run. The weights are reciprocals of the root mean square errors [69]. New variables obtained were used to run a combined ANOVA, corresponded to a nested design with two factors nested within one another [69], where the main-plot factor was weeding treatment, while the subplot factor was monitoring dates. The means of these variables, measured for different times of monitoring and weeding treatments, were separated by Duncan’s test.

On the other hand, the Kruskal–Wallis test was applied to each of the other weed species recorded on each monitoring date because the null hypothesis of normality was not verified. Kruskal–Wallis test is a non-parametric method for testing whether samples originate from the same distribution [70]. Then, the Dunnett test was used to compare the means of weeding treatments in each monitoring date. The ANOVA and the F test for the homogeneity of variances were conducted using Microsoft Excel software (Microsoft Company, Redmond, WA, USA). For the combined ANOVA, Kruskal–Wallis test, Duncan test and Dunnett test, R Studio software (Version 3.6.3) [71] was used.

## 5. Conclusions

This study provides a contribution to the integrated weed management of super high-density olive orchards. It showed that the weeding practices tested had different effects on the floristic composition of weed populations, particularly weeds of different life forms. Mulching has proved to be a viable alternative to soil tillage and chemical control in super high-density olive groves. In particular, as also reported in the literature, mulching with polyethylene and nonwoven tissue were the most effective materials for weed management in the three-year study thanks to their resistance to physical, chemical and biological degradation. However, inorganic mulching, although a more agronomically and environmentally sustainable technique, presents practical problems as they are labor-intensive to apply due to the lack of specialized machinery for the management of plastic and, therefore, costs are higher.

Conversely, this investigation showed that organic mulching with de-oiled pomace, although not as effective as inorganic mulching, proved to be a valid alternative to mowing and chemical weeding. In fact, organic mulches degrade more rapidly than inorganic materials over time; therefore, they have a lower durability. On the other hand, they are easily replaced in super high-density olive orchards thanks to the possibility of using the same machinery used for the distribution of manure and soil amendment in field. Monitoring and sampling conducted several times a year for several years is required to understand when to apply the organic mulch. Further studies could consider the soil seed bank variable and focus on the times and mechanisms of action of de-oiled pomace and, also, on the recovery times of this material in super high-density olive groves. In addition, other organic agro-industrial materials should be investigated.

In conclusion, the use of de-oiled pomace could constitute a useful strategy for sustainable weed management in superhigh-density olive groves, considering these factors:

i.The possibility of mechanizing the distribution of mulch material in super high-density systems and reducing the requirements and costs of labor;ii.The double mode of action of de-oiled pomace (physical and allelochemical);iii.The improvement of soil fertility and some physiological parameters of trees, fruit yield and quality;iv.The opportunity of utilizing and valorizing olive oil extraction waste and, hence, reducing costs for the olive industry by providing a “loop back” from the factory (olive mill) to the crop (olive orchard).

## Figures and Tables

**Figure 1 plants-12-02921-f001:**
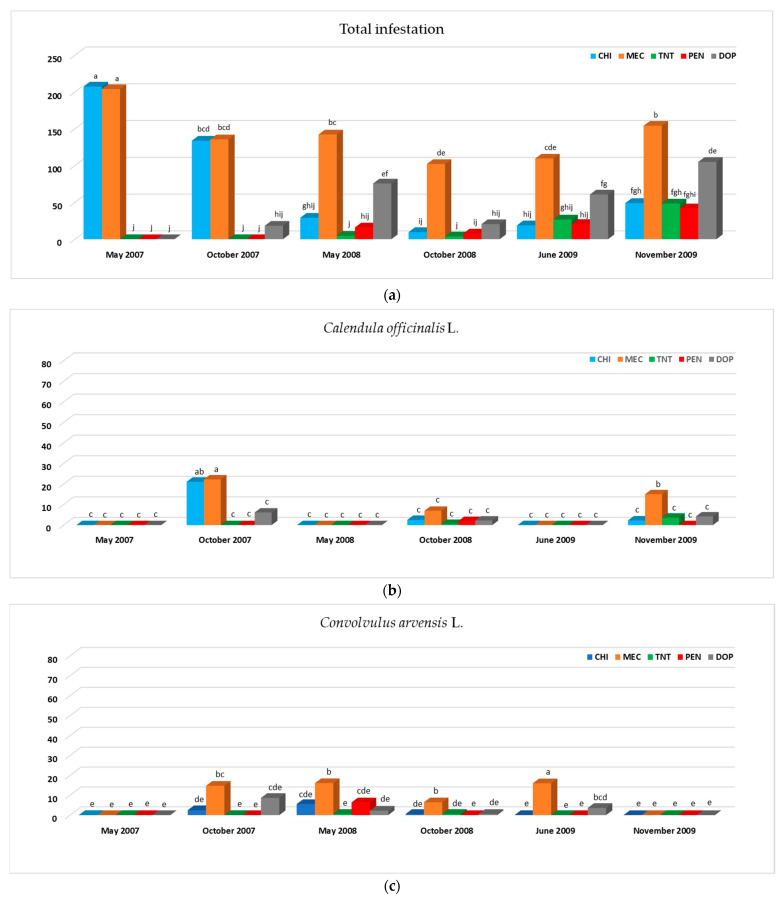
Total infestation (%; **a**) and infestation (%) of *Calendula officinalis* L. (**b**), *Convolvulus arvensis* L. (**c**), *Conyza canadensis* L. (**d**), *Diplotaxis erucoides* L. (**e**), *Medicago polymorpha* (**f**) and *Sonchus oleraceus* L. (**g**) on each monitoring date in relation to different weed managements: mulching with de-oiled pomace (DOP), mulching with nonwoven tissue (TNT) and polyethylene (PEN), chemical weeding (CHI) and mechanical weeding (MEC). Bars followed by different letters are significantly different at a *p*-value of 0.05 (Duncan test).

**Figure 2 plants-12-02921-f002:**
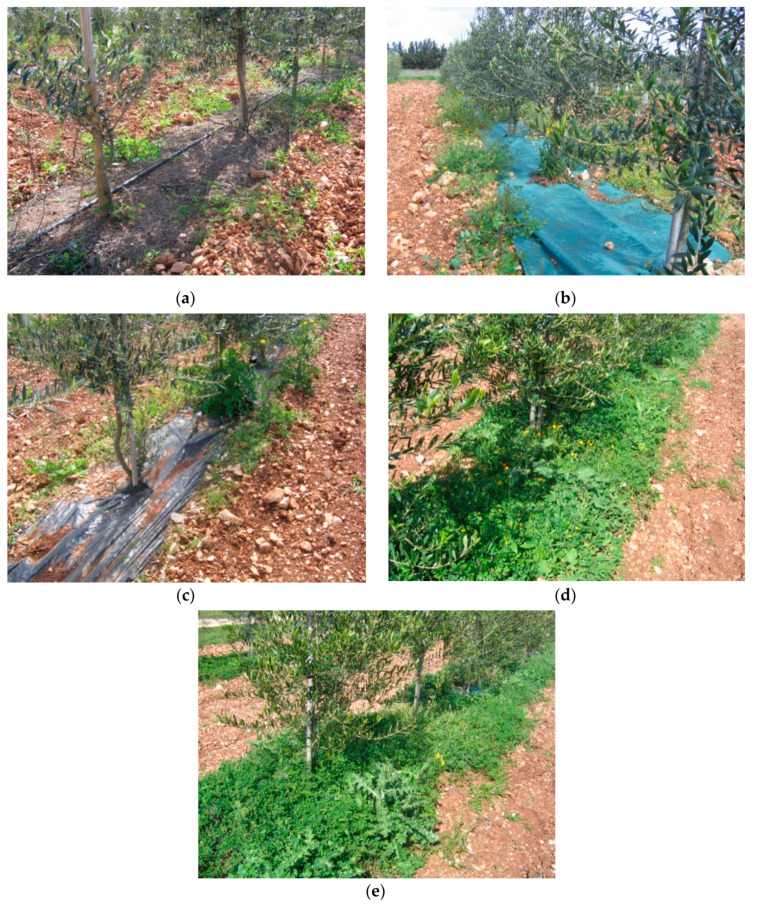
Different weed managements (March 2008): (**a**) mulching with de-oiled pomace (DOP); (**b**) mulching with nonwoven tissue TNT) and (**c**) polyethylene (PEN); (**d**) chemical weeding (CHI); (**e**) mechanical weeding (MEC).

**Table 1 plants-12-02921-t001:** Weed species grouped by life form and detected on each monitoring date.

Life Form	Weed Species	Monitoring Date					
		May 2007	October 2007	May 2008	October 2008	June 2009	November 2009
Hemicryptophytes	*Hirschfeldia incana* L.			x			
	*Lactuca serriola* L.					x	
Cryptophytes	*Convolvulus arvensis* L.		x	x		x	
Therophytes	*Amaranthus retroflexus* L.	x					
	*Anagallis arvensis* L.	x					
	*Avena sterilis* L.						x
	*Calendula officinalis* L.		x				x
	*Chenopodium album* L.	x					
	*Conyza canadensis* L.		x	x	x	x	x
	*Diplotaxis erucoides* L.	x	x				
	*Hordeum murinum* L.						x
	*Lolium rigidum* Gaud.						x
	*Medicago polymorpha* L.		x	x			x
	*Mercurialis annua* L.		x				
	*Senecio vulgaris* L.						x
	*Setaria verticillata* L.	x					
	*Sonchus oleraceus* L.	x	x	x	x	x	
	*Stellaria media* L.						x

**Table 2 plants-12-02921-t002:** *p*-value results and significance of combined ANOVA carried out on weed management and monitoring date for total infestation and weed species detected on two or more monitoring dates.

Variables	Weed Management	Season	Weed Management × Season
	*p*-Value		
Total infestation	2.2 × 10^−16^ ***	2.6 × 10^−10^ ***	2.2 × 10^−16^ ***
*Calendula officinalis* L.	1.5 × 10^−6^ ***	1.3 × 10^−10^ ***	1.5 × 10^−6^ ***
*Convolvulus arvensis* L.	3.9 × 10^−14^ ***	5.7 × 10^−8^ ***	1.4 × 10^−9^ ***
*Conyza canadensis* L.	2.2 × 10^−16^ ***	1.1 × 10^−8^ ***	5.1 × 10^−14^ ***
*Diplotaxis erucoides* L.	6.2 × 10^−8^ ***	2.1 × 10^−8^ ***	6.0 × 10^−9^ ***
*Medicago polymorpha* L.	2.9 × 10^−10^ ***	4.0 × 10^−8^ ***	9.3 × 10^−9^ ***
*Sonchus oleraceus* L.	8.2 × 10^−15^ ***	2.3 × 10^−3^ **	2.3 × 10^−8^ ***

Significant at ** *p*-value ≤ 0.01, *** *p*-value ≤ 0.001. Weed management (df) = 4; monitoring date (df) = 5; weed management ×monitoring date (df) = 20.

**Table 3 plants-12-02921-t003:** Total infestation coefficient (%) and weed species (%) recorded in two or more monitoring dates in relation to different weed managements: mulching with de-oiled pomace (DOP), mulching with nonwoven tissue (TNT) and polyethylene (PEN), chemical weeding (CHI) and mechanical weeding (MEC).

Source of Variation	Total Infestation Coefficient (%)	*Calendula* *officinalis* L.	*Convolvulus arvensis* L.	*Conyza* *canadensis* L.	*Diplotaxis erucoides* L.	*Medicago* *polymorpha* L.	*Sonchus* *oleraceus* L.
CHI	74.62 b	4.32 b	1.44 b	17.28 b	2.87 a	1.07 b	10.54 b
MEC	141.29 a	7.37 a	8.89 a	34.49 a	1.69 a	7.92 a	22.83 a
TNT	13.91 d	0.70 c	0.19 b	11.27 c	0.00 b	0.37 b	1.04 c
PEN	14.38 d	0.62 c	1.07 b	9.64 c	0.01 b	0.52 b	1.74 c
DOP	46.69 c	2.1 bc	2.51 b	10.9 bc	0.19 b	10.19 a	9.18 b
May 2007	82.27 a	0.00 c	0.00 c	14.6 ad	0.11 b	0.11 c	8.89 bc
October 2007	57.56 b	9.89 a	5.17 a	6.56 d	4.67 a	2.18 b	8.73 bc
May 2008	53.50 b	0.00 c	6.17 a	9.63 c	0.34 b	6.99 a	13.90 a
October 2008	28.74 c	2.86 b	1.63 a	13.71 b	0.23 b	0.39 c	5.57 c
June 2009	47.29 b	0.00 c	3.95 b	26.61 b	0.00 b	0.00 c	12.0 ab
November 2009	79.72 a	5.37 b	0.00 c	29.3 ac	0.36 b	14.4 ab	5.34 c

Within each column, data followed by different letters are significantly different at a *p*-value of 0.05 (Duncan test).

**Table 4 plants-12-02921-t004:** Weed species infestation (%) recorded on a single monitoring date in relation to different weed managements: mulching with de-oiled pomace (DOP), mulching with nonwoven tissue (TNT) and polyethylene (PEN), chemical weeding (CHI) and mechanical weeding (MEC).

Monitoring Date	Weed Species	Weed Management				
		CHI	MEC	TNT	PEN	DOP
May 2007	*Amaranthus retroflexus* L.	16.9 a	17.2 a	0.0 b	0.0 b	0.0 b
	*Anagallis arvensis* L.	15.3 a	12.5 a	0.0 b	0.0 b	0.0 b
	*Chenopodium album* L.	41.4 a	51.9 a	0.0 b	0.0 b	0.0 b
	*Setaria verticillata* L.	22.8 ab	23.9 a	0.0 b	0.0 b	0.0 b
October 2007	*Mercurialis annua* L.	18.1 a	10.3 ab	0.0 c	0.0 c	1.1 bc
May 2008	*Hirschfeldia incana* L.	0.0 b	18.9 ac	0.0 b	1.1 b	0.0 bc
October 2008	-	-	-	-	-	-
June 2009	*Lactuca serriola* L.	0.1 a	1.7 a	0.1 a	0.1 a	8.1 a
November 2009	*Avena sterilis* L.	3.6 ab	12.3 a	0.1 b	0.1 b	6.1 a
	*Hordeum murinum* L.	5.6 ab	16.4 a	0.6 b	0.1 b	7.5 ab
	*Lolium rigidum* Gaud.	4.5 abc	13.6 a	0.1 c	0.6 bc	8.1 ab
	*Senecio vulgaris* L.	1.7 ab	6.1 a	0.6 b	0.6 b	1.7 ab
	*Stellaria media* L.	2.6 ab	14.7 a	0.1 b	2.0 b	4.8 ab

Within each row, data followed by different letters are significantly different at a *p*-value of 0.05 (Dunnett test).

## Data Availability

The data supporting the conclusions of this article will be made available by the Authors, without undue reservation.

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
