# Peer review of "Different Weed Managements Influence the Seasonal Floristic Composition in a Super High-Density Olive Orchard"

_plants, 2023, doi:10.3390/plants12162921_

Round 1
Reviewer 1 Report
The structure of the manuscript has to be revised. It makes no sense that "Material and Methods" to come after "Results" and "Discussion"
Author Response
Thanks for the suggestion, but the “Instruction for authors” of this journal suggests the order of paragraphs as follows: 1) Introduction; 2) Results; 3) Discussion; 4) Materials and Methods; 5) Conclusion
Reviewer 2 Report
Comments for the Author
This article focuses on the significance of weed management in Olive orchards. Weed reduced the economic production of the crop and its management without environmental pollution is essential for a sustainable agroecosystem. To achieve this aim author, investigate the impact of integrated weed management systems in this study. Weeds are general control through inorganic pesticides which have a negative impact on groundwater, soil health, and other component of the environment. The authors observed effective control of weeds through an organic mulching approach and these findings are interesting. The overall manuscript is well written and is within the aim and scope of the Journals and therefore I recommend this for publication. However, there is some minor suggestion that can be updated before the publication.
Minor comments:
Abstract:
· Quantitative finding is lacking??
· Abstracts of every good publication end with future recommendations at a comprehensive level which is lacking in this manuscript and therefore suggested to add one line of future recommendation.
Introduction
· Suggested adding the scientific name, Olive
· Novelty: still has scope for improvement.
Result
· This section can be split into more than one section if possible (To improve the readability of the manuscript).
· Avoid repetition of presentation (Give either table or Figure)
Other comments:
· Remove self-citation.
NA
Author Response
- Abstract:
- Quantitative finding is lacking??
Done
- Abstracts of every good publication end with future recommendations at a comprehensive level which is lacking in this manuscript and therefore suggested to add one line of future recommendation.
Done
- Introduction:
- Suggested adding the scientific name, Olive
Done
- Novelty: still has scope for improvement.
Done
- Results
- This section can be split into more than one section if possible (To improve the readability of the manuscript).
Done
- Avoid repetition of presentation (Give either table or Figure)
The figure and tables show data from different statistical analysis and/or different weeds.
- Other comments:
- Remove self-citation.
We strongly reduced
Reviewer 3 Report
See enclosed doc.
There is alot of text on the effects of the different treatments which is often difficult to follow . I suggest that there should be more referral to the figs and tables and less text.

English language edited throughout in enclosed doc.
Author Response
- There is a lot of text on the effects of the different treatments which is often difficult to follow. I suggest that there should be more referral to the figs and tables and less text.
Thanks for your suggestion. We have tried to improve these paragraphs.
- Abstract L 14 : what are TNT and PEN ?
Modified.
- L 18 “Life form” what is meant ?
We have added what it is.
- L 26 Recovery of what ? Weeds, mulch, ?
We have specified.
- Line 91/2 “due to inactive groups, which are resistant to biological attacks” what does this mean ? Degradation also due to radiation effects on plastic, particularly were tree canopy not developed.
Improved.
- L 219 “Controlled more effectively ….” compared to ?
Modified.
- L 488 This sentence does not make sense. Needs to be rewritten to clarify meaning.
We hope it is clearer now.
- L 530 . No seed bank data is supplied in this paper so how can this be a conclusion ?
We have modified this consideration adding the soil seed bank as a variable to consider in future investigation.

Round 2
Reviewer 3 Report
L 481 : re write as : It considers that the same space can be occupied by either a few larger plants or by numerous small plants.
English now improved
Author Response
made
thank you!